# Leaves, Infusion, and Grounds—A Three–Stage Assessment of Element Content in Yerba Mate *(Ilex paraguariensis)* Based on the Dynamic Extraction and Mineralization of Residues

**DOI:** 10.3390/foods13040509

**Published:** 2024-02-06

**Authors:** Anna Różewska, Jędrzej Proch, Przemysław Niedzielski

**Affiliations:** Department of Analytical Chemistry, Faculty of Chemistry, Adam Mickiewicz University, Uniwersytetu Poznańskiego 8, 61-614 Poznań, Poland; annroz2@st.amu.edu.pl (A.R.); przemyslaw.niedzielski@amu.edu.pl (P.N.)

**Keywords:** ICP OES, water extraction, acid mineralization, elemental composition, yerba mate grounds

## Abstract

The more yerba mate infusions that are consumed, the larger the amount of grounds generated. What is more, both the infusion and the residues after brewing remain rich elements. Therefore, a strategy for the three-stage assessment of the element content was presented. A new brewing method was based on dynamic extraction, ensuring both the ease of preparing the infusion and recovering the grounds. In turn, microwave-assisted acid mineralization was used to decompose the leaves and twigs of yerba mate before and after brewing. In total, 30 products were analyzed by ICP OES in three fractions each, i.e., dry yerba mate, infusion, and grounds, to determine up to 25 elements. The elemental composition was considered in terms of the country of origin, type, or composition of yerba mate. The extraction percentages obtained with dynamic extraction were comparable to previously used ultrasound-assisted extraction, as well as data from the literature. The three-stage strategy is a novel approach in yerba mate studies, and it may be a model procedure for the laboratory preparation of yerba mate grounds (waste that can be re-used, e.g., a natural fertilizer).

## 1. Introduction

In recent years, yerba mate has gained popularity beyond its traditional regions (South America), and its consumption has spread to other parts of the world. This increased interest is not only among consumers but also among scientists who are studying the plant from various points of view. This is understandable because *Ilex paraguariensis* is a unique material. Dried leaves and twigs contain many organic substances, i.e., purine alkaloids, polyphenols, and saponins [1,2], as well as those that are rich in elements [1,3,4]. Yerba mate, although frequently compared to tea (*Camellia sinensis*) and named “mate tea” [2], is completely different in terms of the elemental composition as well as the fraction of elements released into the infusion [4,5].

On the one hand, the determination of elements in dry yerba mate and infusions has become more popular. However, it is still niche compared to all the research on yerba mate. According to the Scopus database (Jan 2024), 1003 documents were found under the keyword “yerba mate”, while the keywords “yerba mate elements” contained 49 documents only (less than 4.9%). Due to this, there is a lot of freedom in the procedures for preparing infusions. Although various yerba mate producers recommend using a 1:10 ratio (50 g of leaves and twigs to 500 mL of water at a temperature of 70–80 °C), the authors use different proportions, e.g., 25 g to 200 mL [4], 1 g to 50 mL [6], 0.5 g to 20 mL [7,8], 10 g to 80 mL [9]. Nevertheless, these are examples of the use of the 1:10 ratio, i.e., 1 g to 10 mL [5] and 10 g to 100 mL [9]. When it comes to water temperature, boiling water dominates (100 °C) [6,7,8,9], although this is contrary to the correct procedure for brewing yerba mate. There are also a few studies comparing the content of elements extracted in cold and warm water [5,10]. Recently, ultrasound-assisted extraction [5] was also conducted using both tap and deionized water.

Since it is a local and easily accessible product in the Southern Cone countries of South America, high consumption is culturally determined. Uruguay, Argentina, and Southern Brazil have the highest annual consumption, respectively, at 8–10 kg, 6.5 kg, and 3–5 kg per capita [11,12]. Although the consumption of the brew is varied, it leads to the generation of a large number of grounds. Moreover, this is enlarged as the world’s yerba mate popularity increases. Therefore, these residues are a potential material for further usage. The use of yerba mate grounds was reported as cost–effective and an efficient adsorbent in the removal of metal ions, Pb(II), Cr(III), and Cr(VI), from aqueous solutions [13]. Similar studies were also reported for other materials, e.g., exhausted tea waste as an adsorbent for the removal of Cu and Pb from wastewater [14]. These grounds were used in the development of high–performance lithium–sulfur battery cathodes [15] and poly(lactic acid)–based bionanocomposite films [16]. The recycling of by–products from the production of yerba mate is another promising area for conducting scientific research. This kind of waste was applied as a reinforcing filler of polypropylene (PP) polymer composites [17].

The use of this type of material in fertilization can be more important in reducing waste. We created fertilizers in capsules from yerba mate powder, which was extracted from mills and dried (INYM, Argentina) [18]. Furthermore, yerba mate powder has been used as a matrix for capsule N fertilizers [19]. The research seems to be promising, and yerba mate powder has a chance to exist as an environmentally friendly ingredient of controlled-release fertilizers [20]. Since it is known that yerba mate grounds still contain many organic nutrients, including phenols and flavonoids [21], it can be also rich in elements because water should leach major and essential trace elements partially. However, there are a lack of studies assessing the content of elements at three stages of yerba mate brewing: dry leaves and twigs and infusion and grounds (residues).

In the following research, a strategy for the three–stage assessment of the element content is presented. In this case, a simple and fast procedure of infusion preparation based on dynamic extraction is presented. In turn, microwave–assisted acid mineralization was employed to decompose the leaves and twigs of yerba mate before and after the brewing. It should be emphasized that 30 different yerba mate products were used for this purpose. The element content leached after dynamic extraction was compared and discussed with those obtained with ultrasound–assisted extraction, which was reported earlier [3,5]. This three–stage strategy is supposed to be a model procedure for the laboratory preparation of yerba mate grounds. This approach can be an introductory point in terms of their further usage.

## 2. Materials and Methods

### 2.1. Sampling

Thirty samples of different yerba mate products from eight brands were purchased online (legal distribution in Poland). Samples were originated from Brazil (*n* = 11), Paraguay (*n* = 13), and Argentina (*n* = 6). Most of these samples (*n* = 19) were repackaged in Poland (and distributed under a Polish trademark). According to their kind (type), 16 samples were “con palo” (a mixture of >70% leaves and <30% twigs), and 14 samples were “despalada” (a mixture of >90% leaves and <10% twigs). According to their composition (purity), 12 samples were pure (containing 100% *Ilex paraguariensis*), while 18 samples contained some additives. Full details are shown in the Appendix A.

### 2.2. Reagents and Gases

All solutions and extracts were prepared using high-pure deionized water (>18 MΩ cm resistivity) obtained from the water purification system Milli–Q (Merck Millipore, Darmstadt, Germany). All standard solutions were prepared with commercial ICP calibration standards (Romil, Cambridge, UK). Concentrated nitric acid (65% HNO_3_) was acquired from Merck (Darmstadt, Germany). High–pure argon (N—5.0, purity 99.999%) was purchased from Linde Gaz Polska (Kraków, Poland).

### 2.3. Sample Preparation

All samples were homogenized using an agate laboratory grinder (Pulverisette, Fritsch GmbH, Idar–Oberstein, Germany). Afterward, a three-stage procedure of sample preparation was conducted (Figure 1).

An accurately weighed sample (0.500 ± 0.001 g) of dry yerba mate was mineralized at 180 °C with 5.0 mL of 65% HNO_3_ in closed PTFE containers (55 mL) using the microwave digestion system, Mars 6 (CEM, Matthews, NC, USA). The process was conducted in the following three stages: ramping (20 min), holding (20 min), and cooling (20 min). After mineralization, a sample was transferred to a PP Falcon tube and diluted with deionized water to 10 mL.Parallel to the first stage, an accurately weighed sample (4.00 ± 0.01 g) of dry yerba mate was placed in a polypropylene funnel on filter paper (previously washed with 200 mL of DI water). Deionized water was heated to 80 °C in an electric kettle, Styline TWK8613P (Bosch GmbH, Stuttgart, Germany), after being cleaned (boiled) with 1% (*w*/*v*) citric acid (Merck, Darmstadt, Germany). When water was reached, the temperature was kept (with the “Keep warm” function), and eight portions of water (5 mL each) were dosed on the sample (in total 40 mL). A flowing down extract was collected in a PP Falcon tube and acidified, adding 2–3 drops of 65% HNO_3_. Residues of yerba mate (grounds) were left to the third stage.After the second stage, yerba mate residues were dried in the laboratory dryer at 50 °C. Moreover, the grounds were weighed before and after drying to calculate the weight loss. Afterward, the whole first step of the procedure was repeated.

All samples were stored no longer than a week before analysis (in the laboratory fridge at +4 °C).

Additionally, a moisture analyzer was applied to determine the water content in fresh prepared wet grounds of yerba mate. The moisture analyzer, MA 210.X2.A (Radwag, Radom, Poland) is equipped with a halogen (as a heating module), ensuring a drying temperature of up to 250 °C. The drying method was set in the standard mode, and it was programmed for the temperature (120 °C) and time interval (20 s) for the weight measurement. The process was completed automatically when a constant mass was reached. Five samples were randomly selected, and 2.0–2.5 g of wet grounds was used (taken rapidly after dynamic extraction).

### 2.4. Instrumentation

Three kinds of samples were analyzed using inductively coupled plasma optical emission spectrometry. The instrument Agilent 5110 ICP–OES SVDV (Santa Clara, CA, USA) was used to determine the selected elements. The radio frequency (RF) power was 1.2 kW, while the plasma, nebulizer, and auxiliary gas flow rates were 12, 0.7, and 1.0 L min^−1^, respectively. A synchronous vertical dual view (SVDV) was used (the viewing height for radial plasma observation was 8 mm). The signal was measured in 3 replicates (5 s each). Instrumental quantification limits (IQLs) were ascertained as the 10 standard deviations (SD) of multiple blank measurements (*n* = 10), while the method quantification limits (MQLs) were calculated for each procedure of sample preparation. Those experimental parameters (including emission lines, nm) are presented in the Appendix A. All samples were repeated twice (*n* = 2), and the results were corrected with a blank sample. During method validation, several certified reference materials (CRMs) with an organic sample matrix were used [4]. Since yerba mate does not have CRMs intended for determining the content of elements, and previous studies have shown that tea leaves (INCT–TL–1) have a different composition and percentage of extractability of elements, the method for adding a standard was additionally used. Acceptable recoveries (80–120%) were achieved for most of the elements in the case of CRM analysis (Appendix A), as well as the standard addition method (Appendix A). The propagated uncertainty for the whole analytical process (including sample preparation) was estimated to be below 20% (a coverage factor k = 2 for approximately 95% confidence).

### 2.5. Statistical Analysis

All statistical analyses were performed using Statistica 13.3 (StatSoft, TIBCO Software Inc., Palo Alto, CA, USA), and the probability value (*p* = 0.05) was applied for all statistical tests. The data distribution was determined using the Lilliefors, Shapiro–Wilk, and Kolmogorov–Smirnov tests. The normality of the distribution was rejected for most of the elements (except boron). According to this, a nonparametric test and Spearman’s rank correlation coefficient were used.

## 3. Results and Discussion

### 3.1. Dry Yerba Mate

The general results of collecting the total content of 25 elements in dry yerba mate, infusions, and grounds, as well as the percentages of the content in dry yerba mate, are presented in Table 1. In the case of dry yerba mate, 20 of the 25 elements (except As, Co, Mo, Pb, and Se) were found above the quantification limits (AQLs) in all samples. Co was only found below the quantification limit (BQL) in Sample no. 15, which was Brazilian–origin despalada with additives, while Se was found BQL in three samples (10, 11, 13). In turn, As and Pb were found BQL in 14 and 15 samples, respectively. It is noteworthy that Mo was found to be AQL in 8 samples of dry yerba mate only, and 7 of them contained additives. This might suggest that this element can be located not in the leaves and twigs of Ilex paraguariensis but in any additives such as herbs, aromas, or fruits. Specific results depending on the origin, kind, and composition (purity) of yerba mate are presented in Appendix A.

In general, the following descending order of the total content (as median) can be arranged as follows: K > Ca > Mg > Mn > P > S > Al > Fe > Si > Na > Zn > Sr > B > Rb > Ti > Cu > Ni > other elements (below 1 mg kg^−1^). In the case of the main components (above 100 mg kg^−1^), the only exception was observed for Brazilian yerba mate (Si > Fe). The same order was also reported [4,6], or single exceptions were observed, i.e.*,* Rb > Sr [5] and Cu > Ti [8]. In the case of moderate concentrated elements (1–100 mg kg^−1^), the mutual orders of B, Rb, and Sr (as the median) were characteristic for countries of origin, i.e.*,* Argentina (B > Sr > Rb), Paraguay (Sr > B > Rb) and Brazil (Sr > Rb > B). For Argentinian, Brazilian, and Paraguayan samples, the following orders were also reported: Sr > B [10] and Sr > Rb [7,21]. In the case of the least concentrated elements (below 1 mg kg^−1^), the following general order was observed (as the medians): Se > Cr > Pb > Cd > As > V > Co > Mo (the least concentrated element). However, completely different orders were arranged according to the country of origin, type, and purity (composition). Many exceptions were observed, e.g.*,* Se > Co > Cr (Argentinian and pure yerba mate), Pb > Se > Cr (Paraguayan), as well as Cd > V > Co (Paraguayan and con palo) and As > Cd > V (Brazilian and despalada). These similarities were probably characteristic of investigated products, e.g.*,* most despadala samples were Brazilian, most Paraguayan samples were con palo, and most Argentinian samples were pure yerba mate. The following order can be found in the literature: Cr > Cd > Co [3,8,22].

The upper limits for arsenic (As), cadmium (Cd), and lead (Pb) in yerba mate, expressed as total concentrations, were defined by South American legislation as 0.6, 0.4, and 0.6 mg kg^−1^, respectively [23]. The maximum limit for As did not exceed any of the samples in which AQL was detected (16). By contrast, the amount of Cd significantly exceeded the limit for 16 of the 30 samples (in the concentration range of 0.42–0.56 mg kg^−1^). The maximum limit for Pb was slightly exceeded in two samples (0.61–0.62 mg kg^−1^). Both samples were of the despalada type and contained additives. Nevertheless, many authors have also reported that Cd and Pb exceeded the recommended level [10,21,24,25].

### 3.2. Infusions

In the case of infusions, concentrations were converted to content (mg kg^−1^) for the easier comparison of all results (Table 1). Specific results depending on the origin, kind, and composition (purity) of yerba mate are presented in Appendix A. From twenty-five elements, Mo and V were detected BQL, while 16 elements were found AQL in all 30 infusions. In turn, Cd and Co were determined to be AQL in 24 and 28 samples, respectively. Elements such as As, Cd, Fe, Pb, Se, and Ti were detected AQL in a few samples only. For example, Pb and Ti occurred in only two (18, 21) and four samples (1, 18, 20, 22), respectively. For the whole population (*n* = 30), it is possible to assume the following order (as the median) K > Ca ≥ Mg > P > Mn > S > Na > Si > Al > Zn > B > Rb > Cu > Sr > Ni > Fe > other elements (below 1 mg kg^−1^). In the case of the main components (above 100 mg kg^−1^), it is possible to distinguish the order Ca > Mg and Mg > Ca. The first order was observed for the Brazilian and Paraguayan yerba mate, despalada samples, and those with additives. In turn, Mg > Ca was observed for the Argentinian, con palo, and pure yerba mate. While the order P > Mn > S was observed for most groups, few exceptions were also found for the Argentinian yerba mate (Mn > P > S) as well as for the con palo and pure yerba mate (P > S > Mn). In the case of the moderately concentrated elements (1–100 mg kg^−1^), the mutual orders of Al, Na, and Si (as the median) were characteristic for countries of origin, i.e.*,* Argentina (Al > Na > Si), Paraguay (Si > Na > Al) and Brazil (Na > Si > Al). For Argentinian, Brazilian, and Paraguayan samples, the following orders were also reported: Sr > B [10] and Sr > Rb [7,21]. In the case of the least concentrated elements in infusions (below 1 mg kg^−1^), completely different orders were observed depending on the origin, type, and purity (composition), although the highest selenium content was the same for all of them.

The extraction percentages (as median) were sorted in descending order: K (61%) > Na (58%) > Ni (56%) > Rb (53%) > As (52%) > B (51%) > P (49%) > S (47%) > Cu (46%) ≥ Co (46%) > Se (45%) > Pb (39%) > Cr (38%) > Zn (34%) > Mg (31%) > Si (29%) > Mn (24%) > Al (15%) > Cd (8.4%) > Sr (8.3%) > Ca (7.4%) > Fe (1.0%) > Ti (0.2%). On the one hand, different percentages of water-extractable contents were reported in the literature [4,5,6,7,8,9,10]. On the other hand, it must be noted that most of these authors proposed their own procedures for water extraction (brewing) due to a lack of standardization in preparing infusions. These differences were intensified by the influence of different hot water temperatures (from 70 to 100 °C), as well as different proportions of the dry sample mass to the volume of water. Based on the latest literature, the following ranges (as the min–max) of extractable percentages were established: Al (1–18%), As (18–52%), B (5.8–81%), Ca (0.34–25.6%), Cd (1.8–55%), Co (46–86%), Cr (5.5–73%), Cu (9.6–65%), Fe (0.13–15%), K (9.3–92%), Mg (6.4–74%), Mn (5.5–69%), Mo (5.4–57%), Na (3–84%), Ni (12.2–90%), P (43–72%), Pb (3.3–75%), Rb (53–91%), S (33–59%), Se (24–48%), Sr (1.6–17%), Ti (0.2–3%), V (1–80%), and Zn (6.1–45%). Contrary to this research, Mo and V were detected AQLs in yerba mate infusions by several authors [7,8,9,10]. In turn, Si was determined in yerba mate infusions once; however, an extractable percentage was not calculated [26]. Summarizing all the above data, most percentages (calculated in this study) were within the min–max range. However, the lowest percentages were found in this study for Co, Rb, and Ti, while the highest percentage was for As only. The comparison of these data with the literature clearly indicates that the procedure of dynamic extraction does not affect the percentages of extractable elements, which is similar to those obtained by other authors. This is further proof that the procedure can be successfully used to prepare yerba mate infusion and developed in subsequent scientific works.

In order to evaluate the procedure of dynamic extraction (DYN), five randomly selected samples of yerba mate were prepared according to the previously presented procedure of ultrasound–assisted extraction (USN) [5]. The full results are reported in the Appendix A (Table 2). In both procedures, deionized water at 80 °C was used. We expected that USN may extract a higher content of investigated elements compared to DYN. Therefore, the extractable content by DYN was stated as a relative percentage (%) of the extractable content by USN (considered as 100%). The value was only calculated whenever an element was determined AQL in the same sample. As expected, the lower content was obtained using DYN (60% of USN as a median percentage for all elements). All content expressed as relative percentages (as medians) were arranged in the following order: Ti (78%) > Cd (77%) > S (76%) > As (75%) > B (71%) > Sr (68%) ≥ Ni (68%) > Co (67%) > Ca (66%) > Cr (61%) ≥ Rb (61%) > K (59%) > Mg (57%) ≥ Se (57%) > Zn (55%) ≥ Cu (55%) ≥ Mn (55%) > Al (53%) > P (52%) > Si (37%) > Na (36%) > Fe (35%). Regardless of the extraction procedure used, Mo, Pb, and V were found BQL in all five samples. The losses of extractable content were significantly compensated by the various advantages of DYN over USN. Firstly, the DYN provides greater freedom in the sample weight and dosage of the extractant. In this way, it is possible to fractionate the yerba mate infusion simulating the brewing several times (similar to the three-step brewing by Baran et al. 2019) [4]. Secondly, the total extraction time of DYN is shorter (10–15 min) than USN (30 min). Thirdly, the recovery of yerba mate residues after USN is usually difficult, and separating the infusions from the grounds is sometimes supported with a vortex. Fourthly, working with an ultrasonic cleaner is bothersome and harmful with long–term exposure. Although the main advantage of USN is that the process is carried out automatically, some elements (especially Se) were more frequently extracted AQL using DYN than USN.

### 3.3. Grounds

For the whole population (*n* = 30), fresh wet grounds of yerba mate were rapidly weighed after dynamic extraction (with a previously weighed dry filter paper) before putting them into a laboratory dryer. The ratio between the dry and wet weight (assuming the wet weight as 100%) of yerba mate grounds ranged from 19.0% to 33.6%, with 24.9% as the median value (mean ± SD was 25.9 ± 3.5%). Additionally, five random samples of fresh wet grounds were analyzed in a moisture analyzer (with drying at 120 °C until a constant mass was reached). After 25 min (1500 s), the constant mass (as the median) was obtained for each sample, equaling 24.1% of the initial mass of wet grounds (Figure 2). In turn, the min–max range was 22.7–31.1% while the mean ± SD was 26.4 ± 3.5%. The results show that the water content in fresh grounds of yerba mate is similar regardless of the method of moisture analysis and the number of samples. It also confirms the representativeness of five random samples for the whole population (*n* = 30). What is more, the shape of the drying curve provides information that the water comes only from the wetting of the yerba mate, and no phase changes were observed. In the literature, the initial weight loss of yerba mate residues (starting below 100 °C up to 105 °C) was related to water evaporation (water weakly bonded on the surface of grounds as well as chemically adsorbed) [17,27]. This process was probably observed for the first 20 s during the rapid heating of the moisture analyzer (Figure 2).

In the case of yerba mate grounds, eighteen elements were found AQL in all samples while only Mo was detected BQL. Cd, V, and B were found to be BQL only in one (26), two (26, 27), and three samples (10, 11, 16), respectively. An interesting observation was that samples 26 and 27 are the same brand. In turn, Se and Co were determined to be AQL in 18 and 19 samples, respectively, while Co was more often detected in the grounds of pure yerba mate (15 samples) than with additives (3 samples). As and Pb were rarely found to be AQL, i.e.*,* 4 and 9 samples, respectively. For the whole population (*n* = 30), the following sequence can be arranged (as the median): Ca > K > Mg > Mn > P > S > Al > Fe > Si > Zn > Sr > Na > B > Rb > Ti > Cu > Ni > other elements (below 1 mg kg^−1^). Compared to the dry yerba mate samples, Ca predominated over K for the grounds. In the case of the other main components, the only exception here was Fe > Si for Argentinian yerba mate. It is worth noting that this order is the same as the order for dry yerba mate. The other exceptions were Na > Sr (for Paraguayan, con palo, and pure yerba mate) and Rb > B (for Brazilian, despalada and pure yerba mate). The order Ti > Cu > Ni was identical for all groups and the same as dry yerba mate. For elements with a low content (less than 1 mg kg^−1^), the following order was arranged (as medians): Se > Pb > V > Cr > Cd > As > Co. Regardless of the country of origin, the kind or purity of yerba mate, the Se content was the highest while the Co content was the lowest in the grounds. However, no similar orders were observed for the other least-concentrated elements. Specific results depending on the origin, kind, and composition (purity) of yerba mate are presented in Appendix A.

The content of elements that remained in the grounds was also presented as a percentage of the content in dry yerba mate samples (expressed as 100%). The percentages were arranged in the following descending order: Ti (87%) > Ca (86%) > V (85%) > Sr (80%) > Cd (76%) ≥ Pb (76%) > Al (74%) > Fe (71%) > Mg (66%) > Zn (60%) > Mn (55%) > Cr (53%) > As (51%) ≥ Cu (51%) > S (50%) ≥ Se (50%) > P (48%) > Rb (43%) > B (38%) > Si (36%) > K (35%) > Na (34%) > Ni (29%) > Co (23%). It is noticeable that when comparing infusions and grounds, the sequence is different, e.g.*,* Ti, Fe, Ca, and Sr (poor extractability, high content in grounds) or K, Na, and Ni (good extractability, low content in grounds). Those poorly extracted elements may constitute the mineral base of fertilizers based on yerba mate grounds. An assessment of the element content in yerba mate grounds is difficult due to the lack of research in the literature. On the one hand, the elemental composition of yerba mate was investigated before and after infusion using particle-induced X-ray emissions (PIXEs) [28]. On the other hand, the use of an X-ray-based technique prevents a reliable comparison of results with those obtained via plasma-based spectrometry techniques. Moreover, PIXE and other X-ray-based techniques are used for solid surface analysis and the analysis of infusion, which is not possible. The authors reported the loss of element content, which was established at different levels, i.e.*,* 90% (K and Cl), 50% (Mg and P), and 20% (Mn, Fe, Cu, Zn, and Rb). In the case of Al and Si, a higher content was found after the infusion compared to before (respectively 107% and 164% of the content in dry yerba mate). The authors established that the first 600 mL of water affected the content of K, Mg, Cl, and P (a steep decrease), while higher water temperature favored the extraction of K and Cl only [28]. In turn, we reported that the content extracted by boiling water (100 °C) ranged from 101% (Ca) to 169% (Rb) of the content extracted from hot water (80 °C) [5].

The content of some elements in the grounds was still high and exceeded 1000 mg kg^−1^ (Ca, Mg) or 100 mg kg^−1^ (Mn, P, S, Al, Fe, and Al). Most of them (K, P, S, Mg, and Ca) are important for plant growth in large quantities. Moreover, yerba mate grounds contain other trace elements, such as Mn, Zn, B, and Cu, which are essential but required in smaller amounts [29]. This provides great prospects for the further use of grounds, which are obtained by a simple and quick method to produce natural fertilizers. On one hand, there is a lack of research in the literature on the elemental analysis of yerba mate grounds with which these results can be compared between studies. On the other hand, similar studies have been conducted with coffee grounds for many years [30]. Conducting further research in the area of yerba mate grounds seems very promising and prospective for this reason.

### 3.4. Spearman’s Rank Correlation Test

Since the normality of data distribution was rejected, Spearman’s rank correlation coefficient (r_s_) was used to present the pairwise associations between all yerba mate samples (dry, infusions, and grounds). Almost all elements were chosen except those that were found BQL in all samples, i.e., Mo (each fraction) and V (infusions).

Each fraction of yerba mate (dry, infusion, grounds) was cross–associated. It was also controlled whether the data distribution had no point-cloud plots. The first pair was dry yerba mate (d) and infusion (i) (Appendix A). In this case, two pairs of strong positive correlations (r_s_ ≥ 0.7) were observed for As and Zn, while there were no strong negative correlations (r_s_ ≤ −0.7). Moderate positive correlations (r_s_ ≥ 0.5) were observed for Na, Se, Cu, Fe(d)/B(i), Co, P(d)/Cd(i), Cu(d)/Se(i), Mn(d)/Co(i), Pb(d)/Si(i), Cr, Rb, Pb(d)/As(i), Cd(d)/Zn(i), Ti(d)/Na(i), Ni, and P(d)/Cr(i). In turn, moderate negative correlations (r_s_ ≤ −0.5) were observed only for Sr(d)/Mg(i), Ni(d)/Mg(i), Al(d)/P(i). Other positive and negative correlations were weak or not statistically significant (*p* < 0.05). For the first pair, similar data are described in the literature, e.g., positive correlations were reported for Ni and Zn [5]. The second pair was dry yerba mate (d) and grounds (g) (Appendix A). There was a pair almost fully positively correlated (r_s_ ≥ 0.9): Si (0.95). Strong positive correlations (r_s_ ≥ 0.7) were observed for Ti, Ca, Zn, V, Pb, Sr, Si(d)/B(g), Fe, Al, and Cu. Moderate positive correlations (r_s_ ≥ 0.5) were observed for Si(d)/Cu(g), Co, Se, B(d)/Zn(g), Ca(d)/Al(g), Si(d)/Sr(g), B, Cu(d)/Si(g), V(d)/Ti(g), Cr, P(d)/Rb(g), As, Mn, B(d)/Na(g), Fe(d)/Zn(g), Al(d)/Ca(g), K(d)/B(g), and Cu(d)/Cd(g). In turn, moderate negative correlations (r_s_ ≤ −0.5) were observed only for Se(d)/Fe(g), Na(d)/Co(g), Mg(d)/Rb(g), Ti(d)/Co(g), Co(d)/Se(g), P(d)/B(g), and Na(d)/Ti(g). Other positive and negative correlations were weak or not statistically significant (*p* < 0.05). The third pair included grounds (g) and infusions (i) (Appendix A). There were only two pairs of moderate positive correlations (r_s_ ≥ 0.5), i.e., Pb(g)/Si(i) and Zn(g)/Mg(g), while moderate negative correlations (r_s_ ≤ −0.5) were observed for K, P, Ni(g)/P(i), Mg, Ni(g)/Mg(i), S, Ni(g)/Zn(i), Co(g)/Na(i), Co(g)/Zn(i), Ni(g)/B(i) and Se(g)/Co(i). Other positive and negative correlations were weak or not statistically significant (*p* < 0.05).

As expected, the strongest and moderate correlations (most often positive) were observed between the first two pairs, dry yerba mate and grounds (d–g), as well as dry yerba mate and infusions (d–i). In turn, the most negative correlations were noticed between infusions (i) and grounds (g). For the d–g pair, many correlations were often observed between the same element with the exception of Cd, Mg, and Ni (weak correlations, r_s_ < 0.5), as well as K, Na, P, and S (not statistically significant correlations, *p* < 0.05). However, other moderate associations were observed for the above-mentioned elements, e.g., negative correlations between grounds and infusions (g–i) for K (−0.670), P (−0.634), Mg (−0.621) and S (−0.556) and positive correlations between dry yerba mate and infusion (d–i) for Na (0.693) and Ni (0.519). For the d–i pair, strong and moderate positive correlations were found for As, Zn, Na, Se, Co, Cr, Rb, and Ni.

Additionally, principal component analysis (PCA) was conducted, and two components were described with 99.8% variability of the results for 25 elements (*n* = 2250 of single results for 30 yerba mate products in 3 fractions). Since the two groups of samples were observed (the first: infusions; the second: dry yerba mate and grounds), the results were considered clear and not shown in the text.

## 4. Conclusions

A three-stage strategy was introduced to determine the content of twenty-five elements, including a new procedure of infusion preparation based on dynamic extraction. Evaluating the three different stages of the yerba mate consumption (dry, infusion, and grounds) allowed us to understand this sample matrix better. Basic research is still needed in the area of the elemental composition of yerba mate samples because the procedure for preparing the infusion is lacking. Moreover, there is no certified reference material (CRM) for the elemental analysis of yerba mate (*Ilex paraguariensis*). The development of this kind of CRM could certainly accelerate and force the introduction of a standardized methodology. The procedure of dynamic extraction may be assessed as a fast and efficient extraction method reflecting the natural conditions of yerba mate brewing better than ultrasound-assisted extraction. Comparing these data with the literature, it is clearly indicated that the proposed procedure did not negatively affect the extraction percentages, and other authors obtained similar values. In addition, the main advantage of dynamic extraction is the easy recovery of the extraction residue (grounds). Accordingly, the optimization of this procedure can be extended with the increase in the sample mass and the water volume (without changing the 1:10 ratio). Regardless of the extraction rate, it was proven that the high content of elements still remained in yerba mate grounds. Thus, they are a potential material that should be recycled and used as an environmentally friendly fertilizer. However, further studies should be carried out in this field. The available literature is now sparse, which additionally encourages research in this area. Moreover, the thermogravimetric analysis of wet yerba mate grounds indicates a high similarity in water absorption with this sample matrix.

## Figures and Tables

**Figure 1 foods-13-00509-f001:**
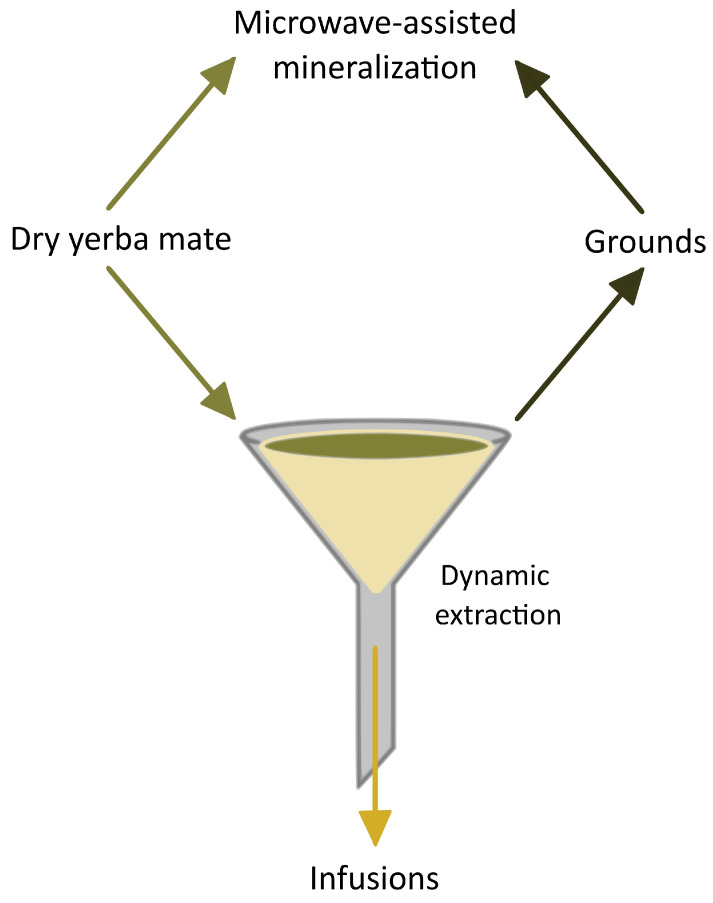
General scheme of the three-stage procedure (details in text).

**Figure 2 foods-13-00509-f002:**
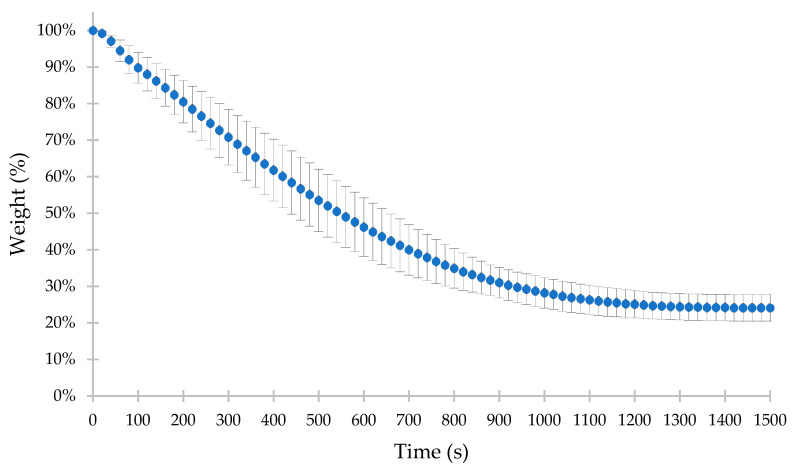
Drying curve of wet yerba mate grounds (as median, min–max) for 5 random samples obtained with the moisture analyzer.

**Table 1 foods-13-00509-t001:** Total content (mg kg^−1^) and percentages (% of the content in dry yerba mate).

Element	Dry	Infusions	Infusions	Grounds	Grounds
	(mg kg^−1^)	(mg kg^−1^)	(%)	(mg kg^−1^)	(%)
	Median {Range} (AQL)	Median {Range} (AQL)	Median {Range} (AQL)	Median {Range} (AQL)	Median {Range} (AQL)
Al	286 {191–375} (30)	42.8 {13.0–71.0} (30)	15 {4.9–25} (30)	212 {137–331} (30)	74 {54–92} (30)
As	0.40 {0.16–0.58} (16)	0.22 {0.09–0.29} (13)	52 {24–87} (13)	0.26 {0.16–0.46} (4)	51 {37–94} (4)
B	35.8 {26.8–55.0} (30)	18.6 {5.03–30.7} (30)	51 {15–85} (30)	16.1 {1.18–28.4} (27)	38 {2.8–79} (27)
Ca	9760 {8630–11200} (30)	728 {269–1460} (30)	7.4 {2.7–16} (30)	8430 {7040–10400} (30)	86 {74–95} (30)
Cd	0.42 {0.19–0.56} (30)	0.04 {0.01–0.06} (24)	8.4 {3.6–17} (24)	0.29 {0.05–0.46} (29)	76 {12–98} (29)
Co	0.29 {0.08 –0.93} (29)	0.13 {0.06–0.38} (28)	46 {8.4–90} (28)	0.10 {0.06–0.21} (11)	23 {8.4–72} (11)
Cr	0.60 {0.40–0.79} (30)	0.22 {0.09–0.50} (30)	38 {15–69} (30)	0.30 {0.15–0.45} (30)	53 {21–69} (30)
Cu	8.42 {5.58–11.1} (30)	3.82 {2.46–5.38} (30)	46 {33–75} (30)	4.05 {1.54–5.52} (30)	51 {24–65} (30)
Fe	221 {92.7–312} (30)	2.29 {1.34–3.69} (28)	1.0 {0.6–2.4} (28)	148 {50.9–238} (30)	71 {36–97} (30)
K	11900 {9900–15300} (30)	7100 {4010–9550} (30)	61 {39–77} (30)	4070 {1760–7830} (30)	35 {16–60} (30)
Mg	2420 {2050–2910} (30)	728 {425–1430} (30)	31 {17–54} (30)	1660 {1100–2010} (30)	66 {45–80} (30)
Mn	1430 {1040–2230} (30)	370 {121–744} (30)	24 {11–41} (30)	814 {411–1330} (30)	55 {31–77} (30)
Mo	0.08 {0.07–0.10} (8)	BQL (0)	N/A	BQL (0)	N/A
Na	85.0 {51.6–102} (30)	48.7 {16.0–78.3} (30)	58 {22–80} (30)	27.8 {12.3–53.0} (30)	34 {14–72} (30)
Ni	4.15 {2.80–5.31} (30)	2.30 {0.89–3.09} (30)	56 {29–77} (30)	1.18 {0.17–2.32} (30)	29 {6.1–44} (30)
P	1030 {884–1120} (30)	497 {269–706} (30)	49 {31–71} (30)	526 {239–712} (30)	48 {23–69} (30)
Pb	0.45 {0.19–0.62} (15)	0.13 {0.10–0.17} (2)	39 {16–62} (2)	0.35 {0.19–0.48} (9)	76 {35–82} (9)
Rb	33.6 {25.9–43.6} (30)	17.5 {11.7–26.3} (30)	53 {36–68} (30)	14.1 {8.26–0.48} (30)	43 {23–63} (30)
S	759 {591–902} (30)	347 {224–541} (30)	47 {32–67} (30)	363 {256–533} (30)	50 {32–66} (30)
Se	0.77 {0.39–1.28} (27)	0.44 {0.27–0.60} (16)	45 {29–88} (16)	0.51 {0.43–0.71} (12)	50 {0.5–90} (12)
Si	153 {87.3–283} (30)	45.7 {21.2–66.2} (30)	29 {15–58} (30)	90.3 {31.1–211} (30)	36 {84–62} (30)
Sr	37.0 {23.8–65.5} (30)	3.16 {1.05–8.16} (30)	8.3 {1.8–26} (30)	28.9 {16.1–48.2} (30)	80 {45–95} (30)
Ti	9.53 {4.07–14.1} (30)	0.02 {0.02–0.07} (4)	0.2 {0.2–0.5} (4)	8.19 {3.71–13.0} (30)	87 {54–99} (30)
V	0.36 {0.10–0.89} (30)	BQL (0)	N/A	0.31 {0.13–0.83} (28)	85 {33 –99} (30)
Zn	71.3 {27.8–117} (30)	24.6 {9.18–39.8} (30)	34 {21–52} (30)	38.6 {16.4–74.2} (30)	60 {30–78} (30)

Range—expressed as content {min–max}; AQL—results above the (method) qualification limit; BQL—results below the (method) quantification limit (provided if AQL = 0); N/A—not applicable.

**Table 2 foods-13-00509-t002:** The comparison of two extraction procedures performed with 5 random samples of yerba mate: dynamic (DYN) and ultrasound–assisted (USN). All results are given as the relative percentage (%) of extractable content (DYN to USN).

Element	Mean ± SD	Median (Min–Max)	AQL (DYN–USN)
Al	52 ± 14	53 (28–66)	5–5
As	78 ± 12	75 (64–98)	5–4
B	70 ± 14	71 (46–85)	5–5
Ca	66 ± 12	66 (47–84)	5–5
Cd	77 ± 17	77 (57–97)	4–3
Co	66 ± 17	71 (38–81)	4–4
Cr	54 ± 18	61 (23–72)	5–5
Cu	53 ± 12	55 (32–65)	5–5
Fe	35 ± 4	35 (29–41)	5–5
K	58 ± 14	59 (34–71)	5–5
Mg	55 ± 14	57 (30–70)	5–5
Mn	51 ± 12	55 (30–63)	5–5
Mo	ND	ND	0
Na	36 ± 5	36 (29–44)	5–5
Ni	62 ± 14	68 (38–77)	5–5
P	54 ± 15	52 (30–70)	5–5
Pb	ND	ND	0
Rb	59 ± 16	61 (32–78)	5–5
S	74 ± 9	76 (63–88)	5–5
Se	57 *	57 *	3–1
Si	39 ± 9	37 (25–49)	5–5
Sr	65 ± 16	68 (38–84)	5–5
Ti	69 ± 29	78 (24–96)	4–4
V	ND	ND	0
Zn	54 ± 12	55 (34–67)	5–5

ND—no data (if AQL = 0); *—single value of the DYN/USN ratio with the dominance of DYN results.

## Data Availability

The data are contained within the article.

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
