# Peer review of "Leaves, Infusion, and Grounds—A Three–Stage Assessment of Element Content in Yerba Mate (Ilex paraguariensis) Based on the Dynamic Extraction and Mineralization of Residues"

_foods, 2024, doi:10.3390/foods13040509_

Round 1
Reviewer 1 Report
Comments and Suggestions for Authors
The article is interesting and well written. the methodology is robust.
The methodology applied to perform a thermogravimetric analysis was not very clear. It is necessary to explain in greater detail, as was done to explain the quantification of elements by ICP.
The presentation of the results and the discussion about the TGA were also not properly explored. So, leave the TGA data and better contextualize the importance of this information for the work or remove the thermal analysis.
Author Response
C1. The article is interesting and well written. the methodology is robust.
R1. We sincerely thank the Reviewer for the positive reception of the manuscript.
C2. The methodology applied to perform a thermogravimetric analysis was not very clear. It is necessary to explain in greater detail, as was done to explain the quantification of elements by ICP.
R2. We agree with the Reviewer and the following methodology has been thoroughly revised:
"Additionally, a moisture analyzer was applied to determine the water content in fresh prepared wet grounds of yerba mate. The moisture analyzer, MA 210.X2.A (Radwag, Radom, Poland), is equipped with a halogen (as a heating module), ensuring the drying temperature up to 250°C. The drying method was set on the standard mode and it was programmed for the temperature (120°C) and time interval (20 s) for the weight measurement. The process was completed automatically when the constant mass was reached. Five samples were randomly selected and 2.0−2.5 g of wet grounds were used (taken rapidly after an dynamic extraction)."
C3. The presentation of the results and the discussion about the TGA were also not properly explored. So, leave the TGA data and better contextualize the importance of this information for the work or remove the thermal analysis.
R3. We sincerely thank the Reviewer for this comment. The term “thermogravimetric analysis” has been used inappropriately and excessively. In fact, TGA is a function of the change in mass and temperature (up to 1000°C), and the discussion was prepared for this kind of analysis. In this work, the procedure was just the determination of water content in fresh wet grounds of yerba mate (done with a moisture analyzer). Unfortunately, we have found no similar data in the literature for yerba mate grounds. The paragraph has been added in the methodology (mentioned above) and the following caption has been revised:
"Figure 2. Drying curve of wet yerba mate grounds (as median, min−max) for 5 random samples obtained with the moisture analyzer."
We hope that the above changes will meet the Reviewer's expectations. If the explanation is not sufficient, this part will be removed.
Reviewer 2 Report
Comments and Suggestions for Authors
Read through the attached file foods-2859823-peer-review-v1-rev1.pdf and correct minor errors.

Author Response
We sincerely thank the Reviewer for the positive reception of our manuscript. We have corrected all indicated errors.
Reviewer 3 Report
Comments and Suggestions for Authors
The article provides a detailed analysis of the elemental composition of yerba mate, examining its content in dry form, infusions, and grounds. While the study covers a broad range of elements and presents a comprehensive methodology, there are a few potential points to consider:
The article provides a detailed account of the methods and results but lacks broader contextualization. It would benefit from a more thorough discussion of the implications of the findings, potential applications, or relevance to existing literature.
The methodology section is quite extensive, which may make it challenging for readers who are not experts in analytical chemistry. A summary or simplification of key steps could enhance accessibility for a broader audience.
The presentation of results, especially in tabular form, could be overwhelming for readers.
While the article mentions the use of statistical tests, the results of these tests are not discussed in detail. Providing insights into the statistical significance of correlations and differences between samples would strengthen the interpretation of the results.
Certain details in the methods section, such as specific instrument models and detailed parameters, may be too technical for a general audience. Consideration should be given to the level of detail necessary for readers to understand and replicate the study.
Addressing these points could enhance the clarity, accessibility, and overall impact of the article.
Comments on the Quality of English Language
Minor editing of English language required
Author Response
The article provides a detailed analysis of the elemental composition of yerba mate, examining its content in dry form, infusions, and grounds. While the study covers a broad range of elements and presents a comprehensive methodology, there are a few potential points to consider:
C1. The article provides a detailed account of the methods and results but lacks broader contextualization. It would benefit from a more thorough discussion of the implications of the findings, potential applications, or relevance to existing literature.
R1. The elemental analysis of yerba mate, both dry and infusions, is still niche (less than 5% of publication about yerba mate), while the grounds are practically not analyzed. It is surprising because exhausted twigs and leaves still remain rich in elements what was confirmed in our research. In turn, daily consumption of yerba mate generates up to 10 kg of grounds per year. Therefore, we decided to meet the need for managing grounds by introducing a three-stage procedure of sample preparation. The dynamic extraction ensures easy and quick preparation of the infusion without loss of element content in the filtrate (comparing to the other extraction methods, Section 3.2) as well as easy recovery of grounds after brewing. In this way, the three–stage procedure can be a model preparation of yerba mate grounds in the future. We see the greatest potential in using this material as fertilizer [1, 2] (similarly to coffee grounds) [3], however grounds have already been used as an adsorbent [4], a substrate for polymer composites [5, 6] or battery cathodes [7]. All above was mentioned several times in the various part of the text and it was summarized in the conclusions.
- Llive L.; Bruno E.; Molina−García A.D.; Schneider Teixeira A. et.al. Biodegradation of Yerba Mate Waste Based Fertilizer Capsules. Efect of Temperature, Polym. Environ. 2019, 27, 1302–1316. DOI:10.1007/s10924–019–01433–y
- Llive, L.M.; Perullini, M.; Santagapita, P.R.; Schneider−Teixeira, A.; Deladino, L. Controlled Release of Fertilizers from Ca(II)−Alginate Matrix Modified by Yerba Mate (Ilex Paraguariensis) Waste. Polym. J. 2020, 138, 109955, DOI:10.1016/j.eurpolymj.2020.109955.
- Junior, M.P.F.; Siqueira, J.C. de; Souza, A. dos R.; Matos, M.P. de; Fia, R. Use of By−Products Generated in the Processing of Coffee Berries: A Review. Coffee Sci. 2023, 18, e182101, doi:10.25186/.v18i.2101.
- Copello, G.J.; Garibotti, R.E.; Varela, F.; Tuttolomondo, M.V.; Diaz, L.E. Exhausted Yerba Mate Leaves (Ilex Paraguariensis) as Biosorbent for the Removal of Metals from Aqueous Solutions. Braz. Chem. Soc. 2011, 22, 790–795, DOI:10.1590/S0103–50532011000400024.
- Hansen B.; Borsoi C.; Dahlem Júnior M. A.; Catto A. L. Thermal and thermo−mechanical properties of polypropylene composites using yerba mate residues as reinforcing filler Crops Prod. 2019, 140,111696. DOI:10.1016/j.indcrop.2019.111696
- Arrieta, M.P.; Peponi, L.; López, D.; Fernández–García, M. Recovery of Yerba Mate (Ilex Paraguariensis) Residue for the Development of PLA–Based Bionanocomposite Films. Crops Prod. 2018, 111, 317–328, DOI:10.1016/j.indcrop.2017.10.042.
- Tesio A. Y.; de Haro Niza J.; Sanchez L. M.; Rodrígez A.; Calballero A. Turning yerba mate waste into high−performance lithium–sulfur battery cathodes, J. Energy Stor. 2023, 67, 107627. DOI:10.1016/j.est.2023.107627
C2. The methodology section is quite extensive, which may make it challenging for readers who are not experts in analytical chemistry. A summary or simplification of key steps could enhance accessibility for a broader audience.
R2. We agree with the Reviewer that the extensive methodology may be a challenge for some readers. However, the introduction of the three-stage methodology was the scope of this research manuscript. We have described all the steps in such a way that readers can open the entire process step by step and we have included only the key parameters for this purpose. Less important parameters were presented in the Supplementary Material (Tables S2-S4). Moreover, the three-stage procedure of sample preparation was visualized (Figure 1) with a broad audience in mind. The methodology has been reduced slightly, however we have added a new paragraph at the request of another Reviewer (details below).
C3. The presentation of results, especially in tabular form, could be overwhelming for readers.
R3. We have presented results of 30 samples at 3 stages for 25 elements, which makes a huge data set. What is more, we decided to add the percentages of element content in infusions and grounds (where 100% was the content found in dry yerba mate). It was impossible to collect the data in a single figure, or even in several figures, without losing the readability of the results. Moreover, the table in the main text makes easy comparison of results between fractions. Summarizing the above, we have decided to keep all results in tables.
C4. While the article mentions the use of statistical tests, the results of these tests are not discussed in detail. Providing insights into the statistical significance of correlations and differences between samples would strengthen the interpretation of the results.
R4. Due to the normality of the data distribution was rejected (this annotation was placed right in Section 2.4), the Spearman’s rank correlation coefficient was applied for 25 elements in 3 fractions, dry yerba mate (d), infusions (i), grounds (g). It was presented 3 pairs, d-i, d-g, g-i, because these relationships are most important in the subject of our manuscript. These results were discussed in the main text while detailed results (3 figures) were placed in Supplementary Material (Figures S1-S3). Additionally, principal component analysis (PCA) were conducted, however the two groups of samples were observed (the first: infusions, the second: dry yerba mate and grounds), the results were considered obvious and not shown in the text.
C5. Certain details in the methods section, such as specific instrument models and detailed parameters, may be too technical for a general audience. Consideration should be given to the level of detail necessary for readers to understand and replicate the study.
R5. With due all respect, we have described all the steps in such a way that a general audience can replicate the entire process step by step. Since all given parameters are crucial to replicate our methodology, we have decided to the following sentence:
"Radio frequency (RF) power was 1.2 kW, while plasma, nebulizer and auxiliary gas flow rates were 12, 0.7, and 1.0 L min−1 respectively."
At the request of another Reviewer, we have extended the methodology with the following paragraph (including the operating parameters of the moisture analyzer):
"Additionally, a moisture analyzer was applied to determine the water content in fresh prepared wet grounds of yerba mate. The moisture analyzer, MA 210.X2.A (Radwag, Radom, Poland), is equipped with a halogen (as a heating module), ensuring the drying temperature up to 250°C. The drying method was set on the standard mode and it was programmed for the temperature (120°C) and time interval (20 s) for the weight measurement. The process was completed automatically when the constant mass was reached. Five samples were randomly selected and 2.0−2.5 g of wet grounds were used (taken rapidly after an dynamic extraction)."
C6. Addressing these points could enhance the clarity, accessibility, and overall impact of the article.
R6. We sincerely thank for all comments. We hope that our explanations will meet your expectations.